# Toxic Metals in Particulate Matter and Health Risks in an E-Waste Dismantling Park and Its Surrounding Areas: Analysis of Three PM Size Groups

**DOI:** 10.3390/ijerph192215383

**Published:** 2022-11-21

**Authors:** Yingjun Wu, Guiying Li, Taicheng An

**Affiliations:** 1Guangdong-Hong Kong-Macao Joint Laboratory for Contaminants Exposure and Health, Guangdong Key Laboratory of Environmental Catalysis and Health Risk Control, Institute of Environmental Health and Pollution Control, Guangdong University of Technology, Guangzhou 510006, China; 2Guangzhou Key Laboratory of Environmental Catalysis and Pollution Control, Key Laboratory of City Cluster Environmental Safety and Green Development of the Ministry of Education, School of Environmental Science and Engineering, Guangdong University of Technology, Guangzhou 510006, China

**Keywords:** heavy metals, size and spatial distribution, particulate matters, health risks, e-waste

## Abstract

Heavy metals generated from e-waste have created serious health risks for residents in e-waste disposal areas. This study assessed how airborne toxic metals from an e-waste dismantling park (EP) influenced surrounding residential areas after e-waste control. PM_2.5_, PM_10_, and total suspended particles (TSP) were sampled from 20 sites, including an EP, residential areas, and an urban site; ten kinds of metals were analyzed using ICP-MS and classified as PM_2.5_, PM_2.5–10_, and PM_10–100_. Results showed that metals at the EP tended to be in coarser particles, while metals from residential areas tended to be in finer particles. A source analysis showed that metals from the EP and residential areas may have different sources. Workers’ cancer and non-cancer risks were higher when exposed to PM_2.5–10_ metals, while residents’ risks were higher when exposed to PM_2.5_ metals. As and Cr were the most strongly associated with cancer risks, while Mn was the most strongly associated with the non-cancer risk. Both workers and residents had cancer risks (>1.0 × 10^−6^), but risks were lower for residents. Therefore, e-waste control can positively affect public health in this area. This study provides a basis for further controlling heavy metal emissions into the atmosphere by e-waste dismantling and encouraging worldwide standardization of e-waste dismantling.

## 1. Introduction

Many metals, including arsenic (As), cadmium (Cd), chromium (Cr), and nickel (Ni), are known carcinogens, and can lead to lung, skin, or bladder cancer [1]. These metals are also ubiquitous in multiple environments, including the atmosphere [2], water bodies [3], and soil [4]. Some non-carcinogenic metal ions, such as copper (Cu) and iron (Fe), can cause oxidative stress and can subsequently induce DNA damage, lipid peroxidation, protein modification, and other effects. These can lead to diseases including cancer, cardiovascular disease, and neurological disorders [5]. Metals can be absorbed through inhalation, dermal contact, and ingestion. Exposure through inhalation is somewhat inevitable, and can lead some metals, such as lead (Pb), to be deposited in the lungs for long periods [6]. These particles can then form insoluble and more toxic compounds, such as lead phosphate or iron–oxygen binding compounds [7]. In addition, as confirmed with one study using rats, inhaled metal-contained particles, such as uranium particles, can directly enter the brain [8]. This highlights the need to study metal exposure through the inhalation pathways. This includes studying both carcinogenic and non-carcinogenic metals in particulate matter (PM_2.5_, PM_10_, and total suspended particles (TSP)).

A growing body of research shows that toxic metal levels in biological samples, such as human blood and urine, are associated with metals from particulate matter [9,10]. Exposure to toxic metals causes different adverse health impacts for young people, such as respiratory symptoms and asthma [4,11,12], abnormal kidney function [13], and low birth weight in newborns [14]. Studies of older human populations have found that exposure to PM_2.5_-bound toxic metals may cause premature death [15]. People living near industrial areas may face a higher risk of exposure to metals, due to atmospheric transmission [1,16]. High metal exposure levels have been reported in studies of several industries, including wire rope production [17], electroplating [18], and informal e-waste dismantling (home-style workshop, original dismantling such as acid leaching, and open fire burning) [19]. Several studies have reported that informal e-waste dismantling has generated serious metallic pollution in the surrounding environment, causing health problems, especially in developing countries [20,21]. Extremely high levels of heavy metals have been detected in atmospheric particulate matter, surface dust, and in soils from e-waste areas [22,23], leading to elevated Pb levels in blood [24].

Metal pollutants such as Pb, Cr, As, Cd, and Ni have been reported at high levels in e-waste dismantling areas in China [25,26]. Although China no longer imports e-waste, there remain large amounts of locally generated e-waste [27]. In southern China, a formal e-waste dismantling park (dismantling equipped with environmental protection devices with licenses) was built in 2016 to replace informal homestyle e-waste dismantling workshops in residential areas [28]. This government action was taken to prevent residents from being exposed to the pollutants emitted from e-waste dismantling. However, one study reported that high concentrations of Pb and Cr were detected near the formal e-waste recycling factory [29]. In addition, a recent study reported that children living near the formal e-waste dismantling area still suffered a carcinogenic health risk from metals in the soil [4].

Many organic pollutants have been found in the atmosphere and soil after the establishment of the government’s formal e-waste dismantling park [30,31]; however, the metal pollution in environmental matrices other than soil has been somewhat limited [4]. Metals in soil generally accumulate over time, while those in the atmosphere are more recently produced. Thus, measuring toxic metals in the atmosphere is critical for evaluating contamination from recently produced metals, after the establishment of the formal e-waste dismantling park. Meanwhile, simultaneously investigating atmospheric metals in the surrounding residential areas of the e-waste dismantling park allows a further evaluation of the impact of formal e-waste on the surrounding area. It is particularly important to study metals of different particulate sizes, because finer particle sizes are associated with greater risks to human health [32].

Given this background, the purpose of this study was to investigate metal levels in particulate matter in residential areas surrounding the formal e-waste dismantling park in southern China. This was completed by collecting particulate matter from 18 sites distributed in communities or villages around the e-waste dismantling town on consecutive days. The typical metals associated with e-waste dismantling activities were analyzed using inductively coupled plasma mass spectrometry (ICP-MS). The metal size distribution was assessed by simultaneously collecting PM_2.5_, PM_10_, and TSP samples from one sampling site. In addition, a sampling site was set in the e-waste dismantling park to facilitate a correlation analysis of metals between the e-waste dismantling park and residential areas. Furthermore, a health risk assessment was conducted to evaluate the inhalation risks for residents and the e-waste dismantlers after the establishment of the formal e-waste dismantling park.

## 2. Materials and Methods

### 2.1. Study Site and Sample Collection

A total of 19 sampling sites, labeled as the EP site and sites S1–S18, were established at the e-waste area and in the town of Guiyu in Guangdong province, as shown in Appendix A. The weather on each sampling day during November 2017 is summarized in Appendix A and was reported in our previous study [31]. The subtropical monsoon climate in the study area is characterized by long summers and short winters and long rainy seasons. The EP site was set in the e-waste dismantling park, near the dismantling workshops. In addition, one site was also set in a non-e-waste area (113°24′18″ N, 23°02′35″ E) to compare the pollutant levels between residential areas that were exposed and not exposed to e-waste. PM_2.5_, PM_10_, and TSP samples were simultaneously collected at each site, using three high-volume samplers at an airflow rate of 1.05 m^3^/min. The samplers were equipped with PM_2.5_, PM_10_, and TSP impactors, respectively (TH1000, Wuhan Tianhong Environmental Protection Industry Co., Ltd., Wuhan, China). Quartz fiber filters (203 mm × 254 mm, 1851-865, GE Whatman, Maidstone, UK) were used in the sampling loader; these were pretreated by baking them at 450 °C for 3 h to remove moisture and organic matter. Each sampling event was approximately 8 h in duration (from 9:00 to 17:00), with the sampler placed at approximately 1.5 m above the ground.

### 2.2. Sample Pretreatments and Instrumental Analysis

The mass concentrations of PM_2.5_, PM_10_, and TSP were determined by calculating the gravimetric differences between the filters before and after sampling at 25 °C and 20% humidity. One-eighth of each entirely membrane-loaded sample was cut into pieces using ceramic scissors. Each sample was then digested with a 10 mL acid solution, using fully enclosed microwave digestion (MARS6, CEM Corporation, Matthews, NC, USA) at 200 °C for 1 h. The digested acid consisted of 5.55% (*v/v*) HNO_3_ (trace metal grade, A509-P212, Thermo Fisher Scientific Inc., Waltham, MA, USA) and 16.75% (*v/v*) HCl (for trace metal analysis (ppb), CFEQ-4-110005-00EA, CNW, ANPEL Laboratory Technologies Inc., Shanghai, China) in ultrapure water (specific resistance ≥ 18.25 M·cm). This metal analysis method was recommended by the Chinese Ministry of Environmental Protection (MEP) [33], and the temperature ramp-up procedure for microwave digestion was summarized in Appendix A. The digested sample solution was diluted with 1% HNO_3_ and was filtered through a 0.45 μm microporous membrane before the analysis. The ICP-MS (7900, Agilent Technology Co., Ltd., Santa Clara, CA, USA) was used to measure the concentrations of ten metals: Fe, Zn (zinc), Cu, Mn (manganese), Pb, V (vanadium), Ni, Cr, As, and Cd. A multi-element calibration standard solution was purchased from Agilent Technology Co., Ltd., Santa Clara, CA, USA (Lot#: 50-069CRY2). The working standard curve was developed using dilutions with 1% (*v/v*) HNO_3_; the correlations exceeded 0.999. In addition, Sc (scandium), In (indium), and Bi (bismuth) (Lot#: 50-024CRY2, Agilent Technology Co., Ltd., Santa Clara, CA, USA) were used as the internal standard, using a peristaltic pump for matrix correction. 

The concentration of each metal in the particulate matter (C_air_, ng/m^3^) was calculated according to Equation (1).
(1)$$C_{\mathrm{air}}=\left(C-C_{0}\right)\times V_{s}\times \frac{n}{v}$$
where C and C_0_ (ng/L) represent the sample and blank metal concentrations’ digestion acid solution, respectively. V_s_ (L) represents the constant volume of the digestion solution; and v represents the sampling air volume at an ambient temperature (m^3^). The n is the digested proportion of the total sample filter area.

### 2.3. Source Apportionment Analysis

A principal component analysis (PCA) and cluster analysis are suitable multi-variate approaches for environmental-based research studies [34]. The PCA is a commonly used model to identify the sources of heavy metals in soil and particulate matter [35,36,37]. The PCA is a dimensionality reduction algorithm that obtains the main factors and provides the weights (loading scores) of heavy metals for each component [38]. The study adopted the first two factors to be retained, and then a varimax rotation was performed.

A cluster analysis is another dimensionality reduction algorithm that obtains the information among metals in the same categories [39]. In this study, systematic cluster analysis was conducted for metals grouping from the same source. In addition, the correlation coefficient measures the strength of the inter-relationship, which in this study, was the possibility of the same source, between two heavy metals [34]. A Spearman correlation analysis was conducted for the non-linear correlation analysis [31].

### 2.4. Human Health Risk Assessment via Inhalation

The United States Environmental Protection Agency’s (US EPA) current methodology for health assessment is described in a previous study [40]. The health risks associated with metals in PM_2.5_ and PM_2.5–10_ (with a dynamic diameter between 2.5 and 10 μm, obtained by subtracting the PM_2.5_ concentration from the PM_10_ concentration) samples were assessed for their ability to be inhaled; metals in PM_10–100_ (with a dynamic diameter between 10 and 100 μm, obtained by subtracting the PM_10_ concentration from the TSP concentration) samples were considered non-inhalable. In this study, the adjusted air concentration (**C_air-adj_**) was used as the exposure concentration (EC) of populations. The **C_air-adj_** of each metal was calculated using Equation (2), obtained from the EPA website accessed on 30 June 2022 (https://www.epa.gov/expobox/exposure-assessment-tools-routes-inhalation). The EC of Cr was adjusted to 0.034 times of the total Cr [41], because the reference concentration (RfC) and inhalation unit risk (IUR) could be obtained only for hexavalent Cr. The metals Cu, Zn, and Fe were excluded from the health risk assessment, because they had no associated RfC and IUR. Previous studies found that workers in the e-waste dismantling park generally worked 10 h a day, 6 days a week [42]. To facilitate comparisons under the same conditions, the same exposure timing and frequency were used for both workers and residents.

The lifetime cancer risk (LCR) values and hazard quotients (HQ) were the representative terms used for the quantitative evaluation. The LCR is the probability of cancer when exposed to the toxicant. An HQ value or hazard index (HI, or the sum of multiple toxicant HQs) of less than 1 indicates there is no threat to a sensitive population [43]. The RfC and IUR values from the United States Integrated Risk Information System (US IRIS) are used to calculate LCR and HI (Equations (3)–(5)).
(2)$$\mathrm{EC}=C_{\mathrm{air}\text{-}\mathrm{adj}}=C_{\mathrm{air}}\times 10^{-3}\times \mathrm{ET}\times \frac{1\;\mathrm{day}}{24\;\mathrm{hour}}\times \mathrm{EF}\times \frac{\mathrm{ED}}{\mathrm{AT}}$$
(3)LCR=IUR×EC
(4)$${\mathrm{HQ}}_{i}=\frac{\mathrm{EC}}{{\mathrm{RfC}}_{i}}$$
(5)$$\mathrm{HI}=\sum {\mathrm{HQ}}_{i}$$
where EC is the exposure concentration (μg/m^3^); and C_air_ is the concentration of metal in the atmospheric environment (ng/m^3^). ET represents the exposure time (hours/day), set at 10 h; EF represents the exposure frequency (days/year), set at 288 days/year; ED represents exposure duration (in years), set at 24 years; and AT represents the average time (day) of exposure, calculated by ED × 365 days for non-carcinogens and 70 years × 365 days/year for carcinogens [44]. The chemical forms of metals in the air used to select IUR_i_ (mg/m^3^) and RfC_i_ ((μg/m^3^)^−1^) are shown in Appendix A.

### 2.5. Statistical Analysis

IBM SPSS Statistics 13.0 software (SPSS Inc., Chicago, IL, USA) was used for the statistical analysis. The geomean value with standard deviation was used to represent particulate matter and its metal concentrations in residential areas. This approach was used because the data were not normally distributed, based on the Shapiro–Wilk test (Appendix A). A Spearman correlation analysis and PCA were used to investigate the sources of metals in particulate matter. The spatial maps were drawn using ArcGIS 10.2. (GeoScene Information Technology Co., Ltd., Beijing, China). The cluster analysis and data display diagrams were created using Origin Pro 2022 (learning version, OriginLab Corporation, Northampton, NC, USA).

### 2.6. Quality Assurance and Quality Control

All sample containers were soaked with 20% (*v/v*) HNO_3_ overnight, rinsed with ultrapure water (>18.25 Ω), and then dried to protect them from environmental pollution. An internal standard was used to correct for matrix interference and drift. Metal concentrations in samples were corrected using the average of three procedure blanks for every batch, which were treated in the same way as the samples. The recovery of spiked metals (50 μg/L) in the blank quartz filters was 81.57–92.91%, and the detailed information was provided in Appendix A. Results were corrected using the average concentration of the background levels in the 7 quartz fiber filters.

## 3. Results and Discussion

### 3.1. PM_2.5_, PM_10_, and TSP Pollution in E-Waste Area 

Table 1 shows the mass concentrations of PM_2.5_, PM_10_, and TSP. The findings of the descriptive analysis show that the PM_2.5_ concentration (mean) was highest at the EP site (131.75 μg/m^3^), followed by the urban site (89.30 μg/m^3^) and residential areas (median: 70.01 μg/m^3^ and geomean: 77.08 ± 30.31 μg/m^3^). However, some residential areas observed very high PM_2.5_ (147.92 μg/m^3^) exposure with a wide range (39.12–147.92 μg/m^3^).

The results of PM_10_ and TSP were a bit different from PM_2.5_. The highest PM_10_ and TSP concentrations were recorded at the EP site (PM_10_: 235.28 μg/m^3^, TSP: 457.42 μg/m^3^), followed by residential areas (geomean: PM_10_: 116.67 μg/m^3^, TSP: 183.78 μg/m^3^), and the urban site (PM_10_: 111.93 μg/m^3^, TSP: 128.47 μg/m^3^). However, like PM_2.5_, the highest PM_10_ concentration was observed in residential areas (182.30 μg/m^3^). Furthermore, the PM_2.5_, PM_10_, and TSP concentrations in all areas exceeded China’s air quality standard (AQS) guidelines (GB 3095-2012) [45], and European Commission AQS (25 μg/m^3^ for PM_2.5_) (https://environment.ec.europa.eu/topics/air/air-quality/eu-air-quality-standards_en, accessed on 30 June 2022) guidelines.

Figure 1 shows the size fraction of particulate matter in the samples from each sampling site in the e-waste area. The three sites with the highest PM_10–100_ concentrations were EP, S11, and S18. The weather may have played a role because it was cloudy on the sampling day (Appendix A). However, there was no relationship between the concentrations at the EP site and the weather; it was a sunny day during sampling. All of these results indicate that PM_10–100_ made up the largest size fraction of PM in this e-waste area, while PM_2.5_, PM_2.5–10_, and PM_10–100_ pollution were all significant. This highlighted the need to further investigate the composition of the particulate matter.

### 3.2. Metal Pollution Profile in PM_2.5_, PM_10_, and TSP

Ten typical metals found in the e-waste area were further analyzed in the PM_2.5_, PM_10_, and TSP samples (Table 1). The highest concentration of metals was observed at the EP site. All samples showed the highest concentration of Fe and the lowest concentration of Cd. Among the three PM sizes, the highest concentration for all metals was observed in the TSP. Appendix A shows each metal concentration at each sampling site in the residential areas. Though the mean concentrations of analyzed metals were higher in EP samples than in residential areas, some metals showed very high concentrations in the residential area. Atmospheric V in the e-waste area of this study (maximum 23.26 ng/m^3^) indicated light pollution, at similar levels as cities in China (17.9 ± 16.5 888 ng/m^3^) [46]. With the exception of V, no other analyzed metal in the TSP was highest in the residential area.

The individual metal concentrations were compared with the national and international guidelines or standards. The metal limit values or guidelines from the current AQS of China (contains Pb, As, and Cd), European Commission (contains Pb, As, Cd, and Mn), and WHO [47] (contains Pb, As, Cd, Ni, and V) are as follows: Pb: 0.5 μg/m^3^, Cd: 5 ng/m^3^, As: 6 ng/m^3^, Mn: 150 ng/m^3^, Ni: 20 ng/m^3^, and V: 1000 ng/m^3^. For samples from the residential area, only As in some sample sites exceeded the above limit, showing As pollution at some residential sites. For the EP site, all the metals in the TSP sample exceeded the guidelines, e.g., As, Cd, Pb, Mn, and Ni were present at levels that were 2.11, 1.24, 2.49, 1.97, and 6.31 times higher than the AQS or WHO limits, respectively. In the PM_10_ sample, Ni (47.87 μg/m^3^) and As (7.21 ng/m^3^) exceeded the limit. Unfortunately, there is no standardization or guidelines for Cr, Zn, Cu, and Fe in atmospheric PM, indicating that the actual metal pollution may be more severe within the EP and residential areas.

Compared to the metal concentrations in samples from an informal e-waste dismantling informal workshop collected for a previous study, before the centralized dismantling of e-waste [22], Cd, Pb, and Zn concentrations in TSP samples at the EP site decreased sharply, from 80 to 4.54 ng/m^3^ (Cd), 4.42 to 1.03 μg/m^3^ (Pb), and 3.32 to 1.37 μg/m^3^ (Zn). Fe concentrations in TSP samples decreased slightly, from 11.49 to 10.81 μg/m^3^. In contrast, Ni, Mn, and Cu concentrations increased from 80 to 111.12 ng/m^3^ (Ni), 160 to 257.03 ng/m^3^ (Mn), and 540 to 749.79 μg/m^3^ (Cu), and do not have limits under China’s AQS. These results indicate that the formalization of centralized e-waste dismantling may help reduce Cd, Pb, and Zn concentrations in TSP from the EP site. 

In addition, the PM_2.5_ bound metal concentrations in residential areas of this study were compared with those of the informal e-waste dismantling period in 2004 and 2012–2013 [48,49], when the informal e-waste workshops were mainly scattered in residential areas [50]. Cu, Zn, Pb, Cr, Ni, and Cd levels in PM_2.5_ decreased to 0.6% (Cr)–84% (Cd) of those in 2004, in which samples were collected on the third floor of the building near the e-waste baking site [48]. However, Mn concentrations remained at similar levels in the residential areas of this study compared to those in 2004. In addition, Pb and Cd also decreased to 17% (Pb) and 21% (Cd) of those in 2012–2013, but Cr and Mn increased to 1.12 (Cr) and 1.27 (Mn) times [49]. Thus, Mn may not be the characteristic metal of e-waste dismantling activities.

Ni and Pb may be the characteristic metals of e-waste dismantling activities. Ni compounds are classified as Class 1 carcinogens by the International Agency for Research on Cancer, meaning that the chemical is a known human carcinogen [51]. In the informal e-waste dismantling period of 2007, Ni in TSP samples ranged from 50 to 150 ng/m^3^ [22], indicating that formalization of e-waste dismantling may not reduce Ni emissions. However, Ni from the EP site did not spread to the residential areas, where the Ni concentration was found to be below the limit in this study. The particularly high concentration of Ni at the EP site may be caused by the specific e-waste type. In another e-waste dismantling area, Qingyuan, China, the sum of Ni concentrations in size-fraction particles was 3.5 to 19 ng/m^3^ [25], which was significantly lower than levels detected in samples from the EP site in this study. Ni comes from the stainless steel and printed circuit boards of e-waste [52]. The mechanical cutting of bulky printed boards may lead to high Ni levels in TSP samples from the EP site, while Ni concentrations in PM_2.5_ samples were lower than the limit. 

Pb presented at the second highest levels, above the AQS limit. Heavy air Pb pollution has been found in e-waste areas across China (maximum of 1000 ng/m^3^ in PM_2.5_ samples) [49] and India (maximum of 2000 ng/m^3^ Pb in PM_2.5_ samples) [53]. Herein, Pb concentrations at the EP site were significantly lower than those from the informal e-waste area (2.5 to 5.8 μg/m^3^) [22], but still exceeded the limit (500 ng/m^3^). Pb is mainly generated from the soldering of the components adhering to the printed circuit boards [23]. With the elimination of lead-acid batteries and use of lead-free soldering in China, Pb may decrease in e-waste in the future. However, lead-acid batteries are still used and dismantled in India, which may be a driver for high Pb pollution in the air in India. As with Ni, Pb in residential areas did not exceed the limit; however, reduced Pb concentrations in TSP indicate that the development of formal e-waste dismantling may reduce Pb emissions [22].

In sum, Cd, Pb, Ni, Zn, Cu, and Fe in PM_2.5_ decreased after e-waste control, but Mn and Cr seemed to remain at similar levels. Formally dismantling and recycling e-waste support reductions in most toxic metals in PM_2.5_ on the e-waste dismantling site and residential area.

### 3.3. Size Fraction of the Toxic Metals in PM_2.5_, PM_2.5–10_, and PM_10–100_


In this study, the simultaneously sampled PM_2.5_, PM_10_, and TSP were further analyzed to determine the size fraction of metals in PM_2.5–10_ and PM_10–100_ categories. This can help evaluate different health effects, because different particle sizes can be deposited in different depths of the lung [54]. Figure 2 shows the size fractions of the metals in the samples from the EP site, the surrounding residential area (geomean), and the urban site. Appendix A show the size fractions for the metals from each site in the residential area. The metal concentrations in TSP samples were set at a baseline 100%; as such, the proportion of each metal in PM_2.5_, PM_2.5–10_, and PM_10–100_ samples was expressed as a percentage of the concentration in TSP.

In samples from the EP site, nine metals (not including Cr) were present at the highest proportions (36.21–74.44%) in the PM_10–100_ samples. Cr was distributed at a similar proportion in PM_2.5–10_ and PM_10–100_ samples (39.81% and 37.45%). Metals distributed in the PM_2.5–10_ samples made up about 14.72–39.81%; and As, V, and Cd were present at a higher proportion in the PM_2.5_ samples (28.30–35.33%) compared to other metals (10.84–22.74%). Compared with the previous study of the informal dismantling e-waste site, the proportions of Cd, Cr, Ni, Pb, Zn, Mn, and As in PM_2.5_ (43.9–108.3%) [48] were much higher than those in this study. This may be because e-waste was previously roasted over an open flame in an informal e-waste dismantling process, so the fine particles may have been generated by those combustion activities [55]. Generally, As, V, Cd, Cu, Ni, Zn, Mn, Pb, and Fe in samples from the EP site were more highly distributed in PM_10–100_, which is considered to be a non-inhalable fraction. However, the coarse particles or dust may cause ingestion risks, due to pulmonary ciliary clearance after inhalation [56].

In samples from the residential area, the proportions of metals in the coarser particles were reduced. Metals in the PM_10–100_ proportion (14.17–45.94%) were lower in the surrounding residential area compared to at the EP site. Otherwise, metals in PM_2.5_ were higher in the surrounding residential area compared to the EP site. As, Cd, and Pb had the highest proportions in the PM_2.5_ samples (56.52%, 55.09%, and 58.32%, respectively), while the other metals had higher proportions in the PM_2.5_ samples (24.65–51.10%) from the residential sites compared to the samples from the EP site. Meanwhile, the metal proportions in PM_2.5–10_ were similar to those in residential areas (19.85–33.80%) and the EP site. Further, metals were distributed at the highest levels in PM_2.5_ samples from the urban site, at between 21.76% and 80.08%. This showed that urban domestic sources emitted mainly finer-sized metal particles. This is consistent with previous research results from the Pearl Delta area of China [40]. Thus, the elevated proportion of metal in finer particles from residential areas indicates the higher domestic sources’ contribution. However, there are also geographic reasons for different ratios of metal distributions in fine particles. For example, higher humidity levels in coastal areas may cause the faster dry settlement of PM_10–100_ [57,58].

Finer particles cause more harm to exposed populations [59]. As such, even if metal concentrations are lower than the guidelines or limits in residential areas, the combined toxicity from multiple pollutants may not be low, due to the limited number of metals analyzed for this study. Metals from e-waste dismantling activities tend to be distributed in coarser particles (PM_10–100_), while metals in residential areas tend to be distributed in finer particles (PM_2.5_ and PM_2.5–10_). The different size distribution of metals may show diverse emission sources in the EP site and residential area.

### 3.4. Source Analysis

The Spearman correlation coefficients among metals are summarized in Appendix A (EP site) and Appendix A (surrounding residential area). The metals’ levels were significantly correlated (*p* < 0.05), indicating that certain paired metals may have the same source [31]. In samples from the EP site, the Spearman correlations of all pairwise comparisons of Cd, Pb, Mn, Ni, Zn, Cu, and Fe were significantly correlated; the Spearman correlation of As and V was also significant. Cr was not significantly correlated with the other metals. Figure 3A shows that the PCA revealed results that were consistent with the Spearman correlation results: Cd, Pb, Mn, Ni, Zn, Cu, and Fe were sorted into one class; As and V were sorted into another class; and Cr was sorted into its own class. This indicates that Cd, Pb, Mn, Ni, Zn, Cu, Fe, and Cr were mainly affected by principal component (PC) 1. As and V were mainly affected by PC 2, but they were also affected by PC 1. The PCA scores of metals with respect to size fraction (in PM_2.5_, PM_2.5–10_, and PM_10–100_) were low (<1) in each PC, which were close to each other (Figure 3A, triangle labels). This indicates that the metals detected in PM_2.5_, PM_2.5–10_, and PM_10–100_ samples from the EP site were likely from the same source, considered as e-waste dismantling activities.

In previous studies, higher Cd, Pb, Mn, Ni, Zn, Cu, Fe, and Cr concentrations were found in the e-waste area [22,48]. However, Wu et. al. reported that As was likely not generated from e-waste dismantling activities, because there was no difference in the As concentration across the distances between dust sampling sites [60]. As and V have been reported as coexisting in drinking water, and can be caused by the corrosion of pipes or from by-products created from disinfecting drinking water [61]. Thus, the higher concentrations of coexisting As and V may have come from corroded metal waste. Cr is the main material in stainless steel [62], which is also present at high levels in e-waste. 

Based on the discussion above, e-waste dismantling was likely the source of PC 1, given that all analyzed metals were related to e-waste material. Metal pipe corrosion may be the source of PC 2; this corrosion did occur in the e-waste dismantling park. However, the three classifications of metals had loadings in PC 1, with a cumulative variance of 86.9%. It indicates that metals detected at the EP site were mainly from e-waste dismantling activities, with the main metal pollutants being Cd, Pb, Mn, Ni, Zn, Cu, and Fe. 

Determining the source of metals in residential areas is quite complicated. The Spearman correlation analysis (Appendix A) found that As, Cd, Pb, and Zn were significantly correlated with each other in pairwise matching (r > 0.50, *p* < 0.01), and Ni was significantly correlated with Cr (r = 0.86). Some other metals were also significantly correlated with each other, but with an r < 0.50. The PCA (Figure 3B) and cluster analysis (Appendix A) were further applied to determine possible sources. PC 1 and PC 2 differed for samples in the surrounding residential area compared to those in the EP site. Metals were not divided into different groups by size fractions; this indicates they were affected by the same sources. The cumulative variances of PCs 1 and 2 were 33.5% and 25.2%, respectively, indicating that PC 1 and PC 2 were the main sources. However, other sources were also important contributors of metals in the residential area. For example, Zn, Pb, Cd, and As were grouped as a class with high loadings on PC 1, but were also clustered as their own class. Thus, Zn, Pb, Cd, and As were likely co-emitted from the same source in the surrounding residential area. Cu was isolated as its own class, indicating that it was not co-emitted with other metals in the residential area. The PCA result (classification of metals) was further confirmed by the cluster analysis, and the Spearman correlation of metals in the same class was significant (*p* < 0.05).

Appendix A shows the spatial distribution of each metal in the study area. The figure also shows that the weak diffusion pattern of the metals extended from the EP site to the surrounding residential area in the downwind direction (northeast). The areas with higher As, Cd, and Zn concentrations (northwest and southeast) covered two roads. Most metals in the city were caused by traffic emissions [63]; and biomass combustion was also an important source of metals, especially Zn, Pb, Cd, and As [64]. However, the Pb levels did not show the same significant trends as seen for Zn, Cd, and As, due to the relatively high Pb concentration at the EP site. As discussed previously, metals from the EP site tended to be in coarser particles, which can weakly diffuse. Though extremely high Pb was from the EP site, limited Pb could diffuse to the surrounding area. Thus, PC 1 identified in the residential area may come from a traffic source. Moreover, Ni and Cr were generally co-emitted, as they are associated with a common group of stainless-steel materials, which differ from the e-waste dismantling source. Mn and Fe were also generally co-emitted, due to the common combination of iron and steel, and from vehicle wear or metal peeling. V was also grouped with Mn and Fe, which are also present in heavy oils [65,66]. All these indicate that these metals in residential areas may be more related to traffic emissions than e-waste.

### 3.5. Inhalation Health Risk Assessment

For adults, Figure 4A,B show the LCR and HQ associated with toxic metals in PM_2.5_ and PM_2.5–10_ samples. Here, the same exposure time and exposure frequency (e-waste dismantling workers’ working periods) were used to assess adult health risk. The LCR of each carcinogenic metal differed between the PM_2.5_ and PM_2.5–10_ samples. At the EP site, the LCRs associated with Cd, Pb, Cr, and Ni were higher for the PM_2.5–10_ samples compared to the PM_2.5_ samples. In contrast, the LCR associated with As was higher in the PM_2.5_ samples compared to the PM_2.5–10_ samples. However, in the surrounding residential areas, the LCRs associated with all five carcinogenic metals were higher in PM_2.5_ samples compared to in PM_2.5–10_ samples. The same trends seen for As, Cd, Cr, and Ni were also seen for the metal HQ. A higher Mn HQ was seen in the PM_2.5–10_ samples, and a similar V HQ was seen in samples from both the EP site and residential sites (Figure 4B). The accumulated LCR and HI in samples from both the EP and residential areas (Figure 4C,D) further indicated that cancer and non-cancer risks for workers may be from the metals in PM_2.5–10_, while risks for residents may be caused by the metals in PM_2.5_.

Among the five carcinogenic metals found at the EP site, the LCR was highest for Cr, followed in descending order by Pb > As > Ni > Cd. The LCR in residential areas was also highest for Cr, followed in descending order by As > Cd ≈ Pb ≈ Ni. Each single carcinogenic metal exceeded the safe threshold of cancer risk (the blue dash in Figure 4A,C, marking the level of 1.0 × 10^−6^ [56]). Exceptions included Cd and Ni at the EP site and residential area sites, and Pb at the residential area sites. However, the accumulated LCR values for the EP site and residential area sites exceeded the safe threshold (Figure 4C). These results indicated that the people working in the EP and living in the surrounding residential areas were exposed to unsafe levels of cancer risk. As and Cr were the most reported carcinogenic metals in the e-waste area. These metals were also reported as the leading causes of cancer risk from soil exposure for children in this area [4].

Figure 4B shows that the HQ of each metal was less than 1, indicating that workers and residents do not appear to have a heightened non-cancer risk from a single metal. Of the six metals, the HQ of Mn was associated with the highest non-cancer risk; it was mainly present in the PM_2.5–10_ samples. The HI did not exceed 1, except for the worker’s HI in the PM_10_ samples (the sum of HI values associated with the metal in the PM_2.5_ and PM_2.5–10_ samples). This indicates that Mn may be associated with the highest non-cancer risk at the EP site. However, many metals were excluded from the health risk assessment given the absence of an RfC or IUR. Furthermore, the analysis did not consider other exposure scenarios, such as indoor exposure. Therefore, the public health risks in this e-waste area still need research attention, especially with respect to cancer risks.

The study has some strengths and limitations. The study provides the first evidence that the formal e-waste control replacing the scattered homestyle e-waste dismantling workshop helped to prevent the residents from exposure to e-waste metal pollution. The study employs three different sizes of segregated PM sampling and the analysis of ten different metals including important metal pollutants. The study includes sampling from different residential sites to overcome the uncertainty of metal exposure in a residential area. Nevertheless, the study also has some limitations. The study did not use the source apportionment models for source identification, which could have led to source misclassification. In addition, the lack of real-time meteorological information, such as wind speed and terrain conditions, also increased the challenge of source identification.

## 4. Conclusions

This study investigated the influence of airborne metals emitted from an e-waste site on surrounding residential areas, in the first year after the introduction of formal e-waste control. Metals emitted from e-waste dismantling activities tended to be distributed in coarser particles, and Pb, Ni, Fe, Mn, Zn, Cu, and Cd were detected in the e-waste site. A size and spatial analysis found that metals from the EP site had a slight impact on the surrounding residential areas. A source analysis further indicated that formal e-waste control actions have helped prevent the further spread of metals from the EP site to surrounding residential areas, likely reducing public health risks. However, exposure to metals in indoor environments may lead to more serious health risks for people, especially in e-waste dismantling workshops. Future studies should analyze indoor airborne metals to generate a more realistic exposure risk level, to identify the source of risk, and to inform mechanisms for reducing associated public health risks.

## Figures and Tables

**Figure 1 ijerph-19-15383-f001:**
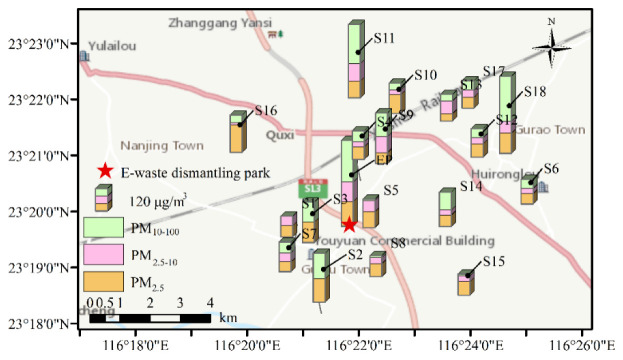
The spatial map of PM_2.5_, PM_2.5–10_, and PM_10–100_ concentrations in the e-waste dismantling park and the surrounding residential area. S1–S18 and EP were labels of sampling points.

**Figure 2 ijerph-19-15383-f002:**
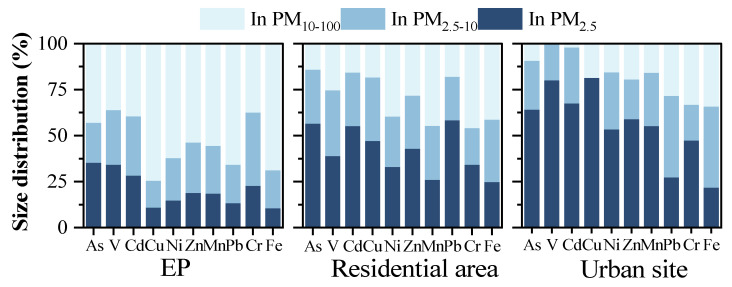
Size distribution of metals in e-waste dismantling park (EP), residential area (geomean of S1~S18), and one urban site.

**Figure 3 ijerph-19-15383-f003:**
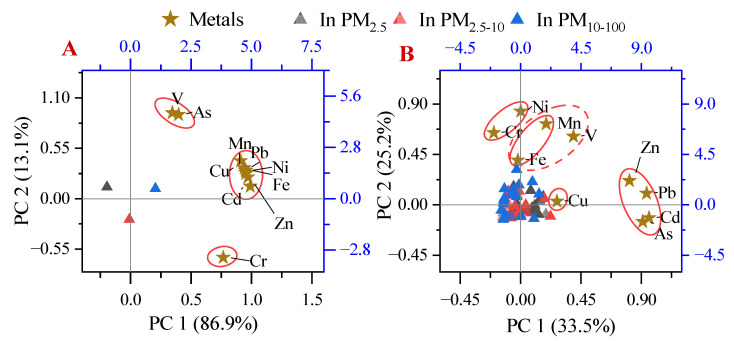
Principal component analysis of the metals in the e-waste dismantling park (**A**) and the surrounding residential area (**B**); principal components and related rotational load are in black, and the scores of the classification are in blue.

**Figure 4 ijerph-19-15383-f004:**
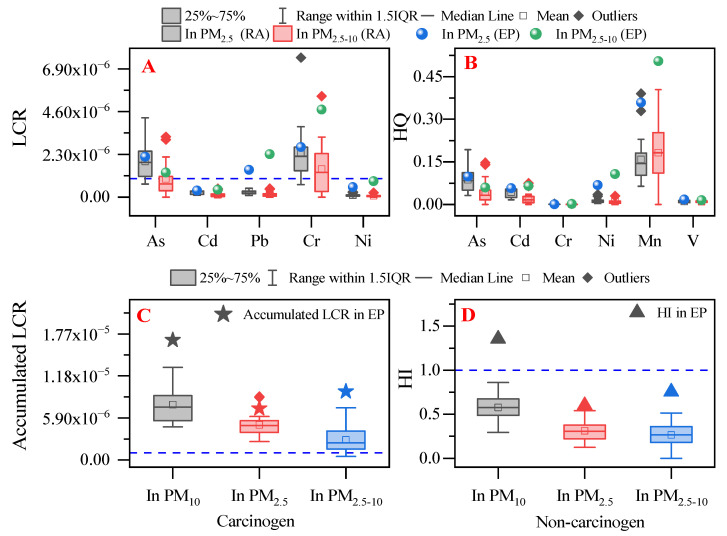
Life cancer risks (LCR) of single metal (**A**), hazards quotient (HQ) of single metal (**B**), the accumulated LCR of five metals (**C**), and hazards index (HI or accumulated HQ) of six metals (**D**) in PM_2.5_ and PM_2.5–10_. EP: e-waste dismantling park; RA: the surrounding residential areas; the blue dash lines: the safe threshold of LCR and HI.

**Table 1 ijerph-19-15383-t001:** Mass concentrations of PM_2.5_, PM_10_, and TSP, and relevant metal concentrations.

Element	EP Site	Surrounding Residential Area		Urban Site
	Mean	Geomean (±SD *)	Median (Range)	Mean
PM_2.5_ (μg/m^3^)	131.75	77.08 (±30.31)	70.01 (39.12–147.92)	89.30
As (ng/m^3^)	4.48	3.51 (±1.99)	3.85 (1.46–8.80)	3.25
Cd (ng/m^3^)	1.75	1.19 (±0.5)	1.47 (0.49–2.08)	0.56
Pb (ng/m^3^)	162.90	27.93 (±12.02)	31.28 (10.09–53.24)	18.52
Cr (ng/m^3^)	8.36	6.36 (±4.68)	6.84 (2.11–23.28)	9.35
Mn (ng/m^3^)	54.54	21.53 (±12.96)	21.90 (9.74–59.33)	33.75
V (ng/m^3^)	5.33	2.82 (±1.33)	2.77 (1.04–5.35)	18.71
Ni (ng/m^3^)	18.59	3.01 (±2.3)	2.69 (1.29–9.73)	9.66
Zn (ng/m^3^)	308.25	95.61 (±60)	101.96 (15.14–242.26)	156.75
Cu (ng/m^3^)	122.11	36.76 (±62.64)	35.85 (3.09–195.76)	71.56
Fe (ng/m^3^)	1607.38	1016.07 (±1.99)	552.92 (244.72–1425.39)	375.89
PM_10_ (μg/m^3^)	235.28	116.67 (±40.68)	103.08 (83.27–182.30)	111.93
As (ng/m^3^)	7.21	5.24 (±2.85)	4.93 (1.85–10.92)	4.60
Cd (ng/m^3^)	3.74	1.72 (±0.76)	1.76 (0.56–3.45)	0.82
Pb (ng/m^3^)	419.38	44.32 (±16.16)	41.95 (19.03–90.86)	48.51
Cr (ng/m^3^)	23.00	10.41 (±3.78)	9.73 (6.85–21.01)	13.17
Mn (ng/m^3^)	131.33	48.89 (±15.62)	47.30 (29.87–85.38)	51.57
V (ng/m^3^)	9.95	5.6 (±1.93)	5.60 (3.06–11.50)	23.36
Ni (ng/m^3^)	47.87	5.27 (±2.12	4.90 (3.54–10.26)	15.27
Zn (ng/m^3^)	758.01	170. (±41)	163.18 (43.51–357.35)	214.39
Cu (ng/m^3^)	287.99	66.81 (±101.45)	68.61 (8.17–368.75)	59.43
Fe (ng/m^3^)	4771.38	1763.88 (±2.85)	1503.17 (986.84–4010.48)	1136.26
TSP (μg/m^3^)	457.42	183.78 (±104.83)	155.42 (106.71–411.34)	128.47
As (ng/m^3^)	12.67	6.39 (±2.64)	7.33 (2.88–11.43)	5.07
Cd (ng/m^3^)	6.19	2.04 (±0.67)	2.24 (0.88–3.37)	0.84
Pb (ng/m^3^)	1224.95	54.28 (±17.59)	56.18 (24.55–92.29)	67.78
Cr (ng/m^3^)	36.77	17.81 (±5.63)	18.42 (11.51–36.01)	19.73
Mn (ng/m^3^)	295.21	86.06 (±44.43)	77.89 (45.88–196.36)	61.25
V (ng/m^3^)	15.60	7.66 (±4.46)	6.72 (4.33–23.26)	21.74
Ni (ng/m^3^)	126.30	8.72 (±3.7)	9.04 (4.46–18.09)	18.10
Zn (ng/m^3^)	1636.70	250.29 (±104.12)	248.36 (113.33–505.69)	266.25
Cu (ng/m^3^)	1126.66	70.11 (±77.67)	75.95 (11.45–315.66)	75.75
Fe (ng/m^3^)	15,309.03	1507.04 (±2.64)	2623.64 (1470.36–11,297.02)	1727.61

* SD: standard deviation.

## Data Availability

Not applicable.

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
