# Peer review of "Toxic Metals in Particulate Matter and Health Risks in an E-Waste Dismantling Park and Its Surrounding Areas: Analysis of Three PM Size Groups"

_ijerph, 2022, doi:10.3390/ijerph192215383_

Round 1
Reviewer 1 Report
- Pay attention to the units of measurement;
- Enter the parameters relating to acid digestion in terms of temperature ramp, time, power, etc; these are important parameters, useful for those who intend to repeat the experiment.
- Line 126: please homogeneize format (for example, Vanadium (V) vs. Ni)
- Table 1: too many digits; please use the adequate number
- Line 193 e sgg. the measure units are micrograms/m3; in table 1 are nanograms/m3..... please check if it is correct
Author Response
Response to the comments for International Journal of Environmental Research and Public Health manuscript ijerph-1968145
Title: Toxic Metals in Particulate Matter and Health Risks in an E-waste Dismantling Park and Its Surrounding Areas: Analysis of Three PM Size Groups
Dear reviewer,
We sincerely thank you for the critical comments and thoughtful suggestions from you that we have used to improve the quality of our manuscript. We have carefully revised the manuscript following your suggestions. The changes in revised manuscript are marked up using the “Track Changes” function in the text and the changes are also highlighted in the text of response. We hope the revised paper will meet your standard. Below you will find our point-by-point responses to your comments:
Point-by-point response:
Question (1): Pay attention to the units of measurement.
Response: Thank you very much for the reviewer’s suggestion. The Eqs. 1 and 2 omitted the unit conversion process and therefore could be easily confused. Following your suggestion, the units have been harmonized in the revised manuscript.
In the revised manuscript, units and the description were revised and marked as following:
(P4, lines 136-138)
Where, C and C0 (ng/L) represent the sample and blank metal concentrations in the digestion acid solution, respectively. Vs (L) represents the constant volume of the digestion solution.
(P4, line 177)
(2)
(P4, lines 181-182)
and Cair is the concentration of metal in the atmospheric environment (ng/m3).
Question (2): Enter the parameters relating to acid digestion in terms of temperature ramp, time, power, etc; these are important parameters, useful for those who intend to repeat the experiment.
Response: Thank you very much for the reviewer’s suggestion. Following your suggestion, the parameters relating to acid digestion procedure were added in Table S2 in the revised Supplementary Materials and shown as following:
Table S2. Digestion program of microwave-assisted acid digestion method
|
Procedure |
Climbing time/min |
Duration /min |
Temperature / ºC |
Power/ W |
|
1 |
15:00 |
5:00 |
170 |
1600 |
|
2 |
08:00 |
8:00 |
180 |
1600 |
|
3 |
08:00 |
15:00 |
200 |
1600 |
|
Total |
|
56:00 |
|
|
In the revised manuscript, the description was revised and marked as following:
(P3, lines 122-124)
This metal analysis method is recommended by the Chinese Ministry of Environmental Protection (MEP) (MEP, 2013) and the temperature ramp-up procedure for microwave digestion was summarized in Table S2.
Question (3): Line 126: please homogenize format (for example, Vanadium (V) vs. Ni).
Response: Thank you very much for the reviewer’s suggestion. The original description in lines 127-128 was because the abbreviated elements (Fe, Cu, Pb, Ni, Cr, As and Cd, in line 32, had already been written in full when it first appeared in the article. But your suggestions have helped the manuscript clearer. Thus, the format of elements was homogenized in the revised manuscript.
In the revised manuscript, the description was revised and marked as following:
(P3, lines 126-128)
ICP-MS (7900, Agilent, US) was used to measure the concentrations of ten metals: Fe, Zn (zinc), Cu, Mn (manganese), Pb, V (vanadium), Ni, Cr, As, and Cd.
Question (4): Table 1: too many digits; please use the adequate number.
Response: Thanks for your critical advice. We deleted the minimum, 25th, 75th and the maximum concentration of surrounding residential area and add range concentrations after median values.
In the revised manuscript, Table 1 was revised and marked as following:
(P6, lines 220-221)
Table 1. Mass concentrations of PM2.5, PM10 and TSP, and relevant metal concentrations
|
Element |
EP site |
Surrounding residential area |
Urban site |
|
|
concentration |
Median concentration (range) |
Geomean concentration ± standard deviation |
concentration |
|
|
PM2.5 (μg/m3) |
131.75 |
70.01 (39.12-147.92) |
77.08 ± 30.31 |
89.30 |
|
As (ng/m3) |
4.48 |
3.85 (1.46-8.80) |
3.51 ± 1.99 |
3.25 |
|
Cd (ng/m3) |
1.75 |
1.47 (0.49-2.08) |
1.19 ± 0.5 |
0.56 |
|
Pb (ng/m3) |
162.90 |
31.28 (10.09-53.24) |
27.93 ± 12.02 |
18.52 |
|
Cr (ng/m3) |
8.36 |
6.84 (2.11-23.28) |
6.36 ± 4.68 |
9.35 |
|
Mn (ng/m3) |
54.54 |
21.90 (9.74-59.33) |
21.53 ± 12.96 |
33.75 |
|
V (ng/m3) |
5.33 |
2.77 (1.04-5.35) |
2.82 ± 1.33 |
18.71 |
|
Ni (ng/m3) |
18.59 |
2.69 (1.29-9.73) |
3.01 ± 2.3 |
9.66 |
|
Zn (ng/m3) |
308.25 |
101.96 (15.14-242.26) |
95.61 ±60 |
156.75 |
|
Cu (ng/m3) |
122.11 |
35.85 (3.09-195.76) |
36.76 ±62.64 |
71.56 |
|
Fe (ng/m3) |
1607.38 |
552.92 (244.72-1425.39) |
1016.07 ±1.99 |
375.89 |
|
PM10 (μg/m3) |
235.28 |
103.08 (83.27-182.30) |
116.67 ± 40.68 |
111.93 |
|
As (ng/m3) |
7.21 |
4.93 (1.85-10.92) |
5.24 ± 2.85 |
4.60 |
|
Cd (ng/m3) |
3.74 |
1.76 (0.56-3.45) |
1.72 ± 0.76 |
0.82 |
|
Pb (ng/m3) |
419.38 |
41.95 (19.03-90.86) |
44.32 ± 16.16 |
48.51 |
|
Cr (ng/m3) |
23.00 |
9.73 (6.85-21.01) |
10.41 ± 3.78 |
13.17 |
|
Mn (ng/m3) |
131.33 |
47.30 (29.87-85.38) |
48.89 ± 15.62 |
51.57 |
|
V (ng/m3) |
9.95 |
5.60 (3.06-11.50) |
5.6 ±1.93 |
23.36 |
|
Ni (ng/m3) |
47.87 |
4.90 (3.54-10.26) |
5.27 ± 2.12 |
15.27 |
|
Zn (ng/m3) |
758.01 |
163.18 (43.51-357.35) |
170. ± 41 |
214.39 |
|
Cu (ng/m3) |
287.99 |
68.61 (8.17-368.75) |
66.81 ± 101.45 |
59.43 |
|
Fe (ng/m3) |
4771.38 |
1503.17 (986.84-4010.48) |
1763.88 ± 2.85 |
1136.26 |
|
TSP (μg/m3) |
457.42 |
155.42 (106.71-411.34) |
183.78 ± 104.83 |
128.47 |
|
As (ng/m3) |
12.67 |
7.33 (2.88-11.43) |
6.39 ± 2.64 |
5.07 |
|
Cd (ng/m3) |
6.19 |
2.24 (0.88-3.37) |
2.04 ± 0.67 |
0.84 |
|
Pb (ng/m3) |
1224.95 |
56.18 (24.55-92.29) |
54.28 ± 17.59 |
67.78 |
|
Cr (ng/m3) |
36.77 |
18.42 (11.51-36.01) |
17.81 ± 5.63 |
19.73 |
|
Mn (ng/m3) |
295.21 |
77.89 (45.88-196.36) |
86.06 ± 44.43 |
61.25 |
|
V (ng/m3) |
15.60 |
6.72 (4.33-23.26) |
7.66 ± 4.46 |
21.74 |
|
Ni (ng/m3) |
126.30 |
9.04 (4.46-18.09) |
8.72 ± 3.7 |
18.10 |
|
Zn (ng/m3) |
1636.70 |
248.36 (113.33-505.69) |
250.29 ± 104.12 |
266.25 |
|
Cu (ng/m3) |
1126.66 |
75.95 (11.45-315.66) |
70.11 ± 77.67 |
75.75 |
|
Fe (ng/m3) |
15309.03 |
2623.64 (1470.36-11297.02) |
1507.04 ± 2.64 |
1727.61 |
Question (5): Line 193 e sgg. the measure units are micrograms/m3; in table 1 are nanograms/m3..... please check if it is correct.
Response: Thank you very much for the reviewer’s suggestion. In the original Table 1, there was a footnote of a: units are μg/m3. However, your suggestion was so helpful to make table 1 display clearer. Thus, we revised the title of Table 1 and removed the footnote. The revised title of Table 1 was as following and the revised Table 1 was shown in the response to question (4).
In the revised manuscript, the title of Table 1 was revised and marked as following:
(P5, line 220)
Table 1. Mass concentrations of PM2.5, PM10 and TSP and relevant metal concentrations.

Reviewer 2 Report
Dear Authors,
The present work presented by you is of great relevance and importance for the health and environmental areas. Since they are showing the importance of monitoring the particulate matter and mainly, identifying and quantifying the types of metals that are present in these particulate materials. The statistical calculations are very good and very well explained, validating the respective results obtained.
However, it was not clear to me, how did you identify the respective sources of emission and contribution of each monitored area without a study of wind roses or emission roses? Considering that for each season of the year, we have different directions, intensities and wind direction.
Best regards.
Author Response
Response to the comments for International Journal of Environmental Research and Public Health manuscript ijerph-1968145
Title: Toxic Metals in Particulate Matter and Health Risks in an E-waste Dismantling Park and Its Surrounding Areas: Analysis of Three PM Size Groups
Dear reviewer,
We sincerely thank you for the critical comments and thoughtful suggestions from you that we have used to improve the quality of our manuscript. We have carefully revised the manuscript following your suggestions. The changes in the revised manuscript are marked up using the “Track Changes” function in the text and the changes are also highlighted in the text of the response. We hope the revised paper will meet your standard. Below you will find our point-by-point responses to your comments:
Point-by-point response:
Question (1): However, it was not clear to me, how did you identify the respective sources of emission and contribution of each monitored area without a study of wind roses or emission roses? Considering that for each season of the year, we have different directions, intensities and wind direction.
Response: Thanks for your critical advice. I’m so sorry that we only recorded the northeast wind as the main wind direction during sampling, and did not record the real-time wind speed and direction during the sampling period. However, according to the meteorological report of the city where the sampling point is located, the northeast wind is the main wind direction, and has been reported in the previous paper (Ge et al., 2020; Yue et al., 2022). Based on that, we did not observe that most of the heavy metals in the particles in the park diffuse in the downwind direction. Therefore, we believe that most of the heavy metals in the particulate matter in the e-waste park would slightly affect the surrounding residential areas. On the contrary, the markable element of traffic emission, such as V, was concentrated in the northwest and southeast area of the e-waste park, where there are mainly two roads in this direction. Therefore, the high-concentration metals in this direction are mainly affected by traffic. As for the different meteorological conditions, we studied the seasonal distribution in another study and did not discuss it in this study. According to your suggestion, we added the limitations of source identification of the study to the revised manuscript. In addition, we will record the real-time wind speed and direction during the sampling period in the future study.
In the revised manuscript, the limitations were added in the text, and marked as follows:
(P13, lines 499-503)
However, there are some limitations of our study. The limited pollutant analysis in this study makes the absence of some source appointment models, such as the positive matrix factorization model. In addition, the lack of real-time meteorological information, such as wind speed and terrain conditions, also increases the challenge of source analysis.
Reference
Ge, X., et al., 2020. Halogenated and organophosphorous flame retardants in surface soils from an e-waste dismantling park and its surrounding area: Distributions, sources, and human health risks. Environ Int. 139, 105741.
Yue, C., et al., 2022. Pollution profiles and human health risk assessment of atmospheric organophosphorus esters in an e-waste dismantling park and its surrounding area. Science of the Total Environment. 806, 151206.

Reviewer 3 Report
PFA

Author Response
Response to the comments for International Journal of Environmental Research and Public Health manuscript ijerph-1968145
Title: Toxic Metals in Particulate Matter and Health Risks in an E-waste Dismantling Park and Its Surrounding Areas: Analysis of Three PM Size Groups
General response:
Dear reviewer,
We sincerely thank you for the critical comments and thoughtful suggestions from you that we have used to improve the quality of our manuscript. We have carefully revised the manuscript, further clarified the logic of source analysis in accordance with your comments. The changes in revised manuscript are marked up using the “Track Changes” function in the text and the changes are also highlighted in the text of response. We hope the revised paper will meet your standard. Below you will find our responses to the total and point-by-point responses to the reviewer’s comments:
Total comment (1): The reviewer has some concerns about the result and the source analysis. The authors have conducted several analyses, and they have tried to incorporate as many results as possible. But, the information in the manuscript needs to be prioritized and focused. The results sometimes don’t follow a sequence and therefore it is hard to associate one result with another. The information looks scattered and all over the place. Thus, the reviewer would advise the authors to maintain a sequence, while writing the result.
Response: We appreciate for your critical advice that help improve our manuscript. Based on your comments and the ideas for solutions provided in the comments, a number of changes have been made in the revised manuscript.
In the revised manuscript, we divided the description of “e-waste area” into “EP site” and “residential areas”, and described the results of residential areas firstly. In addition, we revised the discussion sequence of particulate matter in order from the smallest to largest particle size. The text was revised and marked as follows:
(P5-6, lines 207-233)
Table 1 shows the mass concentrations of PM2.5, PM10 and TSP. All the particulate matter concentrations in the collected samples were followed by TSP > PM10 > PM2.5. Compared the PM2.5 concentration of residential areas with EP site and urban site, the ranks were followed by EP site (131.75 μg/m3) > urban site (89.30 μg/m3) > residential areas (median: 70.01 μg/m3 and geomean: 77.08 ± 30.31 μg/m3). But the highest PM2.5 (147.92 μg/m3) site was in residential areas. Besides, comparing PM2.5 concentration in residential areas with China’s air quality standard (AQS) guidelines (GB 3095-2012) [45], the geomean concentration of PM2.5 exceeded the guideline (75 μg/m3). However, the range of PM2.5 in residential areas was 39.12−147.92 μg/m3, showing that PM2.5 were unevenly distributed in the residential areas. Moreover, compared with the AQS of the European Commission (25 μg/m3 for PM2.5) (https://environment.ec.europa.eu/topics/air/air-quality/eu-air-quality-standards_en), all PM2.5 samples in this study exceeded the standard, showing that PM2.5 pollution was generally high in this region.
The results of PM10 and TSP were different from those of PM2.5 for residential area sites. The ranks of PM10 concentrations were followed by EP site (235.28 μg/m3) > residential areas (geomean: 116.67 μg/m3) > urban site (111.93 μg/m3) > residential areas (median: 103.08 μg/m3). Though the geomean and median PM10 concentrations of residential areas were similar to those of urban sites, the highest PM10 concentration of residential areas (182.30 μg/m3) exceeded the guideline of China’s AQS (150 μg/m3). Further, PM10 concentrations in all samples exceeded the AQS of the European Commission (50 μg/m3 for PM10).
As for TSP pollution, the ranks of concentrations were followed by EP site (457.42 μg/m3) > residential areas (geomean: 183.78 μg/m3 and median: 155.42 μg/m3) > urban site (128.47 μg/m3). Similar to the results of PM10, the highest PM10 concentration at residential area sites exceeded the guideline of China’s AQS (300 μg/m3). The highest PM10 and TSP concentrations were at the EP site among all samples.
(P6, lines 234-235)
However, the different ranks among PM2.5, PM10 and TSP indicate that a specific source of PM10-100 may be present in residential areas.
In the revised manuscript, we reordered result paragraphs in section 3.2 and added articulated sentences to link the two information together. The sentences were added in the text of the revised manuscript and marked as follows:
(P7, lines 259-260)
All regions had higher concentrations of metals at the TSP than PM10 and PM2.5.
(P7, lines 260-261)
For individual metals, Cd concentration was the lowest of all metals, while the Fe concentration was the highest.
(P7, lines 262-265)
Figure S2 shows each metal concentration at each sampling site in the residential areas. Though the concentrations of ten analyzed metals were higher in EP samples than the median and geomean concentration of metals in residential areas, however, some metals’ highest concentrations were seen in the residential area.
(P7, lines 274-275)
The individual metal concentrations were compared with the guidelines or standard of government organization.
(P7, 285-290)
Furthermore, focusing on metals in TSP samples from the EP site, As, Cd, Pb, Mn, and Ni were present at levels that were 2.11, 1.24, 2.49, 1.97, and 6.31 times higher, respectively than the AQS or WHO limits; followed in descending order by Ni > Pb >As > Mn > Cd. However, there is a lack of limit values or guidelines for Cr, Zn, Cu and Fe in atmospheric particulate matter from government organizations or agencies. The actual metal pollution may be more severe within the EP site and residential areas.
(P7, line 310)
Ni and Pb may be the characteristic metals of e-waste dismantling activities.
Total comment (2): For the source analysis, the authors used correlation analysis, PCA and cluster analysis to suggest that some paired metals can arise from a similar source. A valid reference is needed to support that such analysis is adequate to suggest the pollution sources and this could substitute source apportionment models. The authors also suggest that for the residential area the metal sources could be traffic pollution. Thus, how do they justify that formal e-waste dismantling could contribute to the health risk in a residential area? Also, the authors need to provide the absence of proper source appointment as a limitation of the study.
Response: We appreciate for your critical advice that help improve our manuscript. For source analysis, we added a new subsection 2.3 to describe the source appointment analysis and added some references to support that principal component analysis (PCA), correlation analysis and cluster analysis are adequate for such analysis.
We concluded that metals in the surrounding residential area are likely to be more affected by traffic than by e-waste dismantling, thus suggesting that formal e-waste is instrumental in reducing the health risks to the surrounding residents. This conclusion is based on the premise that informal e-waste workshops are mainly scattered in residential areas, especially in residential buildings where the ground floor is used as a dismantling workshop and the upper floors are used as residential areas where all residents, including children, will be directly affected by e-waste dismantling. In contrast, formal e-waste dismantling is done by concentrating all the workshops in the park and using tail gas and waste water treatment equipment and then discharging them. Therefore, when the source of heavy metals in the surrounding residential area is not primarily e-waste, it means that formal e-waste dismantling is beneficial in reducing the health risks to the surrounding population. In the revised manuscript, we strengthen this part of the discussion by adding a comparison of formal and informal e-waste dismantling and also the corresponding health risks. Finally, we added the limitation of absence of proper source appointment in this study as response to question (10).
In the revised manuscript, the descriptions of source analysis methods were added in the revised manuscript and marked as follows:
(P4, lines 140-152)
2.3. Source Apportionment Analysis
Principal component analysis (PCA) and cluster analysis are suitable multivariate approaches for environmental-based research studies [34]. PCA is a commonly used model to identify the sources of heavy metals in soil and particulate matter [35-37]. PCA is a dimensionality reduction algorithm that gets the main factors and provides the weights (loading scores) of heavy metals for each component [38]. The study adopted the first two factors to be retained, and then a varimax rotation was performed.
Cluster analysis is another dimensionality reduction algorithm that gets the information among metals in the same categories [39]. In this study, systematic cluster analysis was conducted for metals grouping from the same source. In addition, the correlation coefficient measures the strength of inter-relationship, in this study, the possibility of the same source, between two heavy metals [34]. Spearman correlation analysis was conducted for non-linear correlation analysis [31].
In the revised manuscript, we revised the comparison of metals with the informal e-waste dismantling period to show the reduction in metals pollution in the text and marked as follows:
(P5, lines 300-309)
In addition, the PM2.5 bound metal concentrations in residential areas of this study were compared with those of the informal e-waste dismantling period in 2004 and 2012-2013 [48, 49], when the informal e-waste workshops were mainly scattered in residential areas [50]. Cu, Zn, Pb, Cr, Ni and Cd levels in PM2.5 decreased to 0.6% (Cr) − 84% (Cd) of those in 2004, in which samples were collected on the third floor of the building near e-waste baking site [48]. However, Mn concentrations remained at similar levels in the residential areas of this study compared to those in 2004. In addition, Pb and Cd also decreased to 17% (Pb) and 21% (Cd) of those in 2012-2013, but Cr and Mn increased to 1.12 (Cr) and 1.27 (Mn) times [49]. Thus, Mn may not be the characteristic metal of e-waste dismantling activities.
In the revised manuscript, the conclusion of the subsection was revised and marked as follows:
(P9, lines 334-337)
In sum, Cd, Pb, Ni, Zn, Cu and Fe in PM2.5 decreased after e-waste control, but Mn and Cr seemed to remain at similar levels. Formally dismantling and recycling e-waste supports reductions in most toxic metals in PM2.5 on e-waste dismantling site and residential area.
Point-by-point response:
Question (1): For ease for the readers who might not be familiar with the acronyms used for metals, authors may provide the full name of the metals in the first place
Response: Thanks for your critical advice. The full name of As, Cd, Cr, Ni, Cu, Fe and Pb were firstly provided in P2, lines 32, 35, and 40. Besides, the full name of Zn, Mn and V were firstly provided in P3, lines127-128. However, to homogenize format of analyzed metals list in P3, lines 127-128, the full name of Zn, Mn and V were provided in brackets after the metal abbreviations. In addition, the full name of Sc, In, and Bi were added in brackets after the metal abbreviations. Follows your suggestion, the full name of the metals have been added when it first appears in the revised manuscript.
In the revised manuscript, the full name of the metals was revised and marked as follows:
(P3, lines 126-128)
ICP-MS (7900, Agilent, US) was used to measure the concentrations of ten metals: Fe, Zn (zinc), Cu, Mn (manganese), Pb, V (vanadium), Ni, Cr, As, and Cd.
(P3, lines 130-132)
In addition, Sc (scandium), In (indium), and Bi (bismuth) (Lot#: 50-024CRY2, Agilent, U.S.) were used as the internal standard, using a peristaltic pump for matrix correction.
Question (2): Some values in the main text and supp. Are provided in exponential, please provide them in absolute numbers.
Response: Thanks for your critical advice. Follows your suggestion, multiple exponential values were revised as the absolute numbers in the revised manuscript.
In the revised manuscript, multiple exponential values revised and marked as follows:
(P5, line 197-198)
All sample containers were soaked with 20% (v/v) HNO3 overnight, rinsed with ultrapure water (>18.25 Ω).
(P5, lines 201-204)
The recovery of spiked metals (50 μg/L) in the blank quartz filters was 81.57%−92.91% and the detailed information was provided in Table S5. Results were corrected using the average concentration of the background levels in the 7 quartz fiber filters.
In the revised Supplementary Materials, Table S5 was revised and marked as follows:
Table S5. The recovery of metals in the blank quartz fiber filters with spiked 50 μg/L metals
|
Element |
1# |
2# |
3# |
Average concentrations |
Standard deviation |
Average recovery |
|
Unit |
μg/L |
% |
||||
|
V |
45.22 |
43.10 |
42.39 |
43.57 |
1.47 |
87.14 |
|
Cr |
44.32 |
42.34 |
39.38 |
42.01 |
2.49 |
84.02 |
|
Mn |
42.93 |
42.12 |
38.67 |
41.24 |
2.26 |
82.47 |
|
Ni |
45.99 |
43.21 |
41.90 |
43.70 |
2.09 |
87.39 |
|
Cu |
45.35 |
42.54 |
41.47 |
43.12 |
2.00 |
86.24 |
|
Zn |
42.42 |
44.77 |
35.17 |
40.78 |
5.00 |
81.57 |
|
As |
47.03 |
46.25 |
46.10 |
46.46 |
0.50 |
92.91 |
|
Cd |
48.73 |
47.67 |
47.66 |
48.02 |
0.61 |
96.04 |
|
Pb |
46.15 |
45.17 |
45.17 |
45.49 |
0.57 |
90.99 |
Question (3): Please describe the abbreviations EC, RfC and IUR in the first place
Response: Thanks for your critical advice. Follows your suggestion, the description of EC was added in the revised manuscript. However, the description of RfC and IUR have been appeared previously.
In the revised manuscript, the description of EC was revised and marked as follows:
(P4, lines 160-161)
In this study, the adjusted air concentration (Cair-adj) was used as the exposure concentration (EC) of populations. The Cair-adj of each metal was calculated using Eq. 2.
In the revised manuscript, the description of RfC and IUR was marked as follows:
(P4, lines 164 and 165)
The reference concentration (RfC) and inhalation unit risk (IUR) could be obtained only for hexavalent Cr.
Question (4): In table 1, it needs to be written specifically what does values under EP site and Urban site represent. Also, why do the authors provide detailed descriptions only for the residential site?
Response: Thanks for your critical advice. We’re sorry that only one sample site was set for e-waste dismantling park (EP site) and the urban site, respectively. Thus, only one series of data was available for EP site and Urban site. Relevant descriptions have been mentioned in the text. As for residential sites, 18 sampling sites were set, which were surrounded the EP site. Thus, we used median and geomean concentration to describe the data of residential sites. However, too many detailed values were shown and we deleted the minimum, 25th, 75th and the maximum concentration of surrounding residential area and added range concentrations after median values.
In the revised manuscript, Table 1 was revised and marked as follows:
(P5, lines 220-221)
Table 1. Mass concentrations of PM2.5, PM10 and TSP, and relevant metal concentrations
|
Element |
EP site |
Surrounding residential area |
Urban site |
|
|
concentration |
Median concentration (range) |
Geomean concentration ± standard deviation |
concentration |
|
|
PM2.5 (μg/m3) |
131.75 |
70.01 (39.12−147.92) |
77.08 ± 30.31 |
89.30 |
|
As (ng/m3) |
4.48 |
3.85 (1.46−8.80) |
3.51 ± 1.99 |
3.25 |
|
Cd (ng/m3) |
1.75 |
1.47 (0.49−2.08) |
1.19 ± 0.5 |
0.56 |
|
Pb (ng/m3) |
162.90 |
31.28 (10.09−53.24) |
27.93 ± 12.02 |
18.52 |
|
Cr (ng/m3) |
8.36 |
6.84 (2.11−23.28) |
6.36 ± 4.68 |
9.35 |
|
Mn (ng/m3) |
54.54 |
21.90 (9.74−59.33) |
21.53 ± 12.96 |
33.75 |
|
V (ng/m3) |
5.33 |
2.77 (1.04−5.35) |
2.82 ± 1.33 |
18.71 |
|
Ni (ng/m3) |
18.59 |
2.69 (1.29−9.73) |
3.01 ± 2.3 |
9.66 |
|
Zn (ng/m3) |
308.25 |
101.96 (15.14−242.26) |
95.61 ±60 |
156.75 |
|
Cu (ng/m3) |
122.11 |
35.85 (3.09−195.76) |
36.76 ±62.64 |
71.56 |
|
Fe (ng/m3) |
1607.38 |
552.92 (244.72−1425.39) |
1016.07 ±1.99 |
375.89 |
|
PM10 (μg/m3) |
235.28 |
103.08 (83.27−182.30) |
116.67 ± 40.68 |
111.93 |
|
As (ng/m3) |
7.21 |
4.93 (1.85−10.92) |
5.24 ± 2.85 |
4.60 |
|
Cd (ng/m3) |
3.74 |
1.76 (0.56−3.45) |
1.72 ± 0.76 |
0.82 |
|
Pb (ng/m3) |
419.38 |
41.95 (19.03−90.86) |
44.32 ± 16.16 |
48.51 |
|
Cr (ng/m3) |
23.00 |
9.73 (6.85−21.01) |
10.41 ± 3.78 |
13.17 |
|
Mn (ng/m3) |
131.33 |
47.30 (29.87−85.38) |
48.89 ± 15.62 |
51.57 |
|
V (ng/m3) |
9.95 |
5.60 (3.06−11.50) |
5.6 ±1.93 |
23.36 |
|
Ni (ng/m3) |
47.87 |
4.90 (3.54−10.26) |
5.27 ± 2.12 |
15.27 |
|
Zn (ng/m3) |
758.01 |
163.18 (43.51−357.35) |
170. ± 41 |
214.39 |
|
Cu (ng/m3) |
287.99 |
68.61 (8.17−368.75) |
66.81 ± 101.45 |
59.43 |
|
Fe (ng/m3) |
4771.38 |
1503.17 (986.84−4010.48) |
1763.88 ± 2.85 |
1136.26 |
|
TSP (μg/m3) |
457.42 |
155.42 (106.71−411.34) |
183.78 ± 104.83 |
128.47 |
|
As (ng/m3) |
12.67 |
7.33 (2.88−11.43) |
6.39 ± 2.64 |
5.07 |
|
Cd (ng/m3) |
6.19 |
2.24 (0.88−3.37) |
2.04 ± 0.67 |
0.84 |
|
Pb (ng/m3) |
1224.95 |
56.18 (24.55−92.29) |
54.28 ± 17.59 |
67.78 |
|
Cr (ng/m3) |
36.77 |
18.42 (11.51−36.01) |
17.81 ± 5.63 |
19.73 |
|
Mn (ng/m3) |
295.21 |
77.89 (45.88−196.36) |
86.06 ± 44.43 |
61.25 |
|
V (ng/m3) |
15.60 |
6.72 (4.33−23.26) |
7.66 ± 4.46 |
21.74 |
|
Ni (ng/m3) |
126.30 |
9.04 (4.46−18.09) |
8.72 ± 3.7 |
18.10 |
|
Zn (ng/m3) |
1636.70 |
248.36 (113.33−505.69) |
250.29 ± 104.12 |
266.25 |
|
Cu (ng/m3) |
1126.66 |
75.95 (11.45−315.66) |
70.11 ± 77.67 |
75.75 |
|
Fe (ng/m3) |
15309.03 |
2623.64 (1470.36−11297.02) |
1507.04 ± 2.64 |
1727.61 |
Question (5): Line no. 209-210, The values provided in the bracket are inappropriate as it does not say the values differ by this much percentage, instead, it is the absolute value. Therefore, either provide with what %age they differ from each other or remove the values.
Response: Thank you very much for the reviewer’s suggestion. I’m sorry for my mistakes and the values provided in the bracket are for urban site. I agreed with your comment. Follows your suggestion, the values provided in the bracket were removed in the revised manuscript.
Question (6): Line no. 231-234, Why are the values coming from different columns? Also, when you mention the range, please mention the metals with the values as well.
Response: Thanks for your critical advice. We concluded the EP site and residential area sites as e-waste area. However, it might confuse readers. Thus, we divided the metals ranges descriptions into EP site and residential area sites in the revised manuscript. In addition, the concentration ranges were added with metals in the revised manuscript following your suggestions.
In the revised manuscript, the metals ranges descriptions were revised and marked as follows:
(P7, lines 253-261)
The metal concentrations at EP site were as follows: for PM2.5, the range was from 1.75 ng/m3 (Cd) to 1607.38 ng/m3 (Fe); for PM10, the range was from 3.74 ng/m3 (Cd) to 4771.38 ng/m3 (Fe); and for TSP, the range was from 6.19 ng/m3 (Cd) to 15309.03 ng/m3 (Fe). The ranges of metals median concentrations in residential area sites were from 1.47 ng/m3 (Cd) to 552.92 ng/m3 (Fe), from 1.76 ng/m3 (Cd) to 1503.17 ng/m3 (Fe) and from 2.24 ng/m3 (Cd) to 2623.64 ng/m3 (Fe) in PM2.5, PM10 and TSP, respectively. All regions had higher concentrations of metals at the TSP than PM10 and PM2.5. For individual metals, Cd concentration was the lowest of all metals, while the Fe concentration was the highest.
Question (7): Line no. 256-257, “These results indicated that Pb, As, Cd, Mn, and Ni in the samples, especially TSP, from the EP site, was heavily polluted” looks incomplete.
Response: Thanks for your critical advice. I apologize for not expressing the conclusion of this subsection correctly. In the revised manuscript, we revised the conclusion of the subsection in the text.
In the revised manuscript, the conclusion of the subsection was revised and marked as follows:
(P8, lines 282-290)
These results show that TSP and PM10 in EP site are polluted by several metals (Pb, As, Cd, Mn and Ni for TSP; As and Ni for PM10), while all of TSP, PM10 and PM2.5 in residential areas are mainly polluted by As. Furthermore, focusing on metals in TSP samples from the EP site, As, Cd, Pb, Mn, and Ni were present at levels that were 2.11, 1.24, 2.49, 1.97, and 6.31 times higher, respectively than the AQS or WHO limits; followed in descending order by Ni > Pb >As > Mn > Cd. However, there is a lack of limit values or guidelines for Cr, Zn, Cu and Fe in atmospheric particulate matter from government organizations or agencies. The actual metal pollution may be more severe within the EP site and residential areas.
Question (8): Line no. 256-257, the use of the term “impacted” is irrelevant and must be replaced.
Response: Thanks for your suggestion. I’m so sorry that we didn’t find the term “impacted” in Line no. 256-257 of the original manuscript. But the terms “impacted” were used in P9, lines 362-364 and P10, line 394 of the original manuscript. Based on that, we replaced the term “impacted” to “affected” in the revised manuscript according to literature (Anaman et al., 2022).
In the revised manuscript, the term “impacted” was replaced to “affected”, and marked as follows:
(P10, lines 396-398)
This indicates that Cd, Pb, Mn, Ni, Zn, Cu, Fe, and Cr were mainly affected by principal component (PC) 1. As and V were mainly affected by PC 2, but were also affected by PC 1.
(P11, lines 427-428)
Metals were not divided into different groups by size fractions; this indicates they were affected by the same sources.
Question (9): Fig. 4, why EP is shown as a point and RA as a box plot?
Response: Thanks for your critical advice. The main reason is that the number of sampling site in e-waste dismantling park and residential areas are different. One sampling site was set in e-waste dismantling park (EP site) and 18 sampling sites were set in residential areas. Thus, only one series of data was obtained for EP site. In Fig. 4, we compared the lifetime cancer risks (LCR) and non-cancer risks (HQ and HI) of people acted in EP site and residential areas sites. Thus, 18 results of residential areas sites were expressed as box plot and one result of EP site was expressed as a point.
Question (10): Provide the strength and limitations in a separate paragraph.
Response: Thanks for your critical advice. Follows your suggestion, the strength and limitations were added in the revised manuscript.
In the revised manuscript, the strength and limitations were added in the text, and marked as follows:
(P13, lines 499-507)
For the first time, this study reveals that the formal e-waste control replaced the scattered home-style e-waste dismantling workshop, helping prevent residents exposed to e-waste metal pollution directly. Several sampling sites have been set up around the e-waste park to assess the impact of heavy metals on airborne particulate matter generated in the e-waste park on the surrounding residential areas. However, there are some limitations of our study. The limited pollutant analysis in this study makes the absence of some source appointment models, such as the positive matrix factorization model. In addition, the lack of real-time meteorological information, such as wind speed and terrain conditions, also increases the challenge of source analysis.
Reference
Anaman, R., et al., 2022. Identifying sources and transport routes of heavy metals in soil with different land uses around a smelting site by GIS based PCA and PMF. Science of the Total Environment. 823.
Deng, W. J., et al., 2006. Atmospheric levels and cytotoxicity of PAHs and heavy metals in TSP and PM2.5 at an electronic waste recycling site in southeast China. Atmospheric Environment. 40, 6945-6955.
Kumar, P., Fulekar, M. H., 2019. Multivariate and statistical approaches for the evaluation of heavy metals pollution at e-waste dumping sites. Sn Applied Sciences. 1.
Lu, X., et al., 2010. Multivariate statistical analysis of heavy metals in street dust of Baoji, NW China. J Hazard Mater. 173, 744-9.
Taner, S., et al., 2013. Fine particulate matter in the indoor air of barbeque restaurants: elemental compositions, sources and health risks. Sci Total Environ. 454-455, 79-87.
Wong, M. H., et al., 2007. Export of toxic chemicals - a review of the case of uncontrolled electronic-waste recycling. Environ Pollut. 149, 131-40.
Yue, C., et al., 2022. Pollution profiles and human health risk assessment of atmospheric organophosphorus esters in an e-waste dismantling park and its surrounding area. Science of the Total Environment. 806, 151206.
Zhai, Y., et al., 2014. Source identification and potential ecological risk assessment of heavy metals in PM2.5 from Changsha. Sci Total Environ. 493, 109-15.
Zhang, Y., et al., 2018. A systemic ecological risk assessment based on spatial distribution and source apportionment in the abandoned lead acid battery plant zone, China. J Hazard Mater. 354, 170-179.
Zheng, X. B., et al., 2016. Ambient Air Heavy Metals in PM2.5 and Potential Human Health Risk Assessment in an Informal Electronic-Waste Recycling Site of China. Aerosol and Air Quality Research. 16, 388-397.

Round 2
Reviewer 3 Report
PFA

Author Response
Response to the comments for International Journal of Environmental Research and Public Health manuscript ijerph-1968145-revised version-track
Title: Toxic Metals in Particulate Matter and Health Risks in an E-waste Dismantling Park and Its Surrounding Areas: Analysis of Three PM Size Groups
Dear Editor and Reviewers,
Thanks for providing us with this great opportunity to submit a revised version of our manuscript. We are grateful to the reviewers for their suggestions on modification in the result description. We have adopted all the rewriting suggestions and further explained one of them. The changes in the revised manuscript are marked up using the “Track Changes” function in the text and the changes are also highlighted in the text of the response. We hope this revised manuscript has addressed your concerns, and look forward to hearing from you. Below you will find our responses to the total and point-by-point responses to the reviewer’s comments:
Point-by-point
Comment (1): Rewrite section “3.1. PM2.5, PM10, and TSP Pollution in E-waste Area” as-
Table 1 shows the mass concentrations of PM2.5, PM10 and TSP. The findings of the descriptive analysis show that PM2.5 concentration (mean) was highest in the EP site (131.75 μg/m3) followed by the urban site (89.30 μg/m3) and residential areas (median: 70.01 μg/m3 and geomean: 77.08 ± 30.31 μg/m3). However, some residential areas observed very high PM2.5 (147.92 μg/m3) exposure with a wide range (39.12−147.92 μg/m3).
The results of PM10 and TSP were a bit different from PM2.5. The highest PM10 and TSP concentrations were recorded in the EP site (PM10;235.28 μg/m3, TSP; 457.42 μg/m3), followed by residential areas (geomean: PM10;116.67 μg/m3, TSP; 183.78 μg/m3) and urban site (PM10;111.93 μg/m3, TSP;128.47 μg/m3). However, like PM2.5, the highest PM10 concentration was observed in residential areas (182.30 μg/m3). Besides, the PM2.5, PM10 and TSP concentration in all areas exceeded China’s air quality standard (AQS) (PM2.5:75 μg/m3, PM10:150 μg/m3, TSP:300 μg/m3) [45], and European Commission AQS (PM2.5:25 μg/m3, PM10: 50 μg/m3) (https://environment.ec.europa.eu/topics/air/air-quality/eu-air-quality-standards_en) guidelines.
Figure 1 shows the size fraction of PM in the samples from each sampling site in the e-waste area. The three sites with the highest PM10-100 concentrations were EP, S11, and S18. The weather may have played a role because it was cloudy on the sampling day (Table S1). However, there was no relationship between the concentrations at the EP site and the weather; it was a sunny day during sampling. All these results indicate that PM10-100 makes up the largest size fraction of PM in this e-waste area, however, PM2.5, PM10 and PM10-100 pollution are all significant. This highlighted the need to further investigate the composition of the particulate matter.
Response: We appreciate your suggestions that help improve our manuscript. In the revised manuscript, we rewrote section 3.1. as per your suggestions. The changes could be found in P5-P6, lines No. 207-242 in the revised manuscript.
Comment (2): Rewrite section “3.2. Metal Pollution Profile in PM2.5, PM10, and TSP 871” as-
Ten typical metals found in the e-waste area were further analyzed in the PM2.5, PM10, and TSP samples (Table 1). The highest concentration of metals was observed in the EP site. All samples showed the highest concentration of Fe and the lowest concentration of Cd. Among the three PM sizes, the highest concentration for all metals was observed in TSP. Figure S2 shows each metal concentration at each sampling site in the residential areas. Though the mean concentrations of analyzed metals were higher in EP samples than in residential areas, however, some metals show very high concentrations in the residential area. Atmospheric V in the e-waste area of this study (maximum 23.26 ng/m3) indicated light pollution, at similar levels as cities in China (17.9 ± 16.5 888 ng/m3) [46]. With the exception of V, no other analyzed metal in TSP was highest in the residential area.
The individual metal concentrations were compared with the national and international guidelines or standards. The metals limit values or guidelines from the current AQS of China (contains Pb, As and Cd), European Commission (contains Pb, As, Cd and Mn) and WHO [47] (contains Pb, As, Cd, Ni and V) are as followed: Pb: 0.5 μg/m3, Cd: 5 ng/m3, As: 6 ng/m3, Mn: 150 ng/m3, Ni: 20 ng/m3 and V: 1000 ng/m3. For samples from the residential area, only As in some sample sites exceeded the above limit, showing the problem of As pollution in some residential sites. For the EP site, all the metals in the TSP sample exceeded the guidelines, For eg. As, Cd, Pb, Mn, and Ni were present at levels that were 2.11, 1.24, 2.49, 1.97, and 6.31 times higher than the AQS or WHO limits. In the PM10 sample, Ni (47.87 μg/m3) and As (7.21 ng/m3) exceeded the limit. Unfortunately, there is no standardization or guidelines for Cr, Zn, Cu and Fe in atmospheric PM, indicating that the actual metal pollution may be more severe within the EP site and residential areas.
From line no. 911-1082 as it is.
Response: We appreciate your suggestions that help improve our manuscript. In the revised manuscript, we rewrote section 3.2. as per your suggestions. The changes could be found in P10-11, lines No. 283-350 in the revised manuscript.
Comment (3): Rewrite the section “Strengths and limitations” as-
The study has some strengths and limitations: The study provides the first evidence that the formal e-waste control replacing the scattered home-style e-waste dismantling workshop, helped to prevent the residents from exposure to e-waste metal pollution. The study employs three different sizes of segregated PM sampling and the analysis of ten different metals that includes important metal pollutants. The study includes sampling from different residential sites to overcome the uncertainty of metal exposure in a residential area. Besides, the study has some limitations. The study didn’t use the source apportionment models for source identification which could have led to source misclassification. In addition, the lack of real-time meteorological information, such as wind speed and terrain conditions, also increased the challenge of source identification.
Response: We appreciate your suggestions that help improve our manuscript. In the revised manuscript, we rewrote “Strengths and limitations” as your suggestions. The changes could be found in P17, lines No. 584-602 in the revised manuscript.
Comment (4): Modify the Table headings as suggested.
Response: Thanks for your advice. In the revised manuscript, the headings of Table 1 were revised as follows:
Table 1. Mass concentrations of PM2.5, PM10 and TSP, and relevant metal concentrations
|
Element |
EP site |
Surrounding residential area |
Urban site |
|
|
Mean |
Geomean (± SD) |
Median (range) |
Mean |
|
|
PM2.5 (μg/m3) |
131.75 |
77.08 (± 30.31) |
70.01 (39.12−147.92) |
89.30 |
|
As (ng/m3) |
4.48 |
3.51 (± 1.99) |
3.85 (1.46−8.80) |
3.25 |
|
Cd (ng/m3) |
1.75 |
1.19 (± 0.5) |
1.47 (0.49−2.08) |
0.56 |
|
Pb (ng/m3) |
162.90 |
27.93 (± 12.02) |
31.28 (10.09−53.24) |
18.52 |
|
Cr (ng/m3) |
8.36 |
6.36 (± 4.68) |
6.84 (2.11−23.28) |
9.35 |
|
Mn (ng/m3) |
54.54 |
21.53 (± 12.96) |
21.90 (9.74−59.33) |
33.75 |
|
V (ng/m3) |
5.33 |
2.82 (± 1.33) |
2.77 (1.04−5.35) |
18.71 |
|
Ni (ng/m3) |
18.59 |
3.01 (± 2.3) |
2.69 (1.29−9.73) |
9.66 |
|
Zn (ng/m3) |
308.25 |
95.61 (± 60) |
101.96 (15.14−242.26) |
156.75 |
|
Cu (ng/m3) |
122.11 |
36.76 (± 62.64) |
35.85 (3.09−195.76) |
71.56 |
|
Fe (ng/m3) |
1607.38 |
1016.07 (± 1.99) |
552.92 (244.72−1425.39) |
375.89 |
|
PM10 (μg/m3) |
235.28 |
116.67 (± 40.68) |
103.08 (83.27−182.30) |
111.93 |
|
As (ng/m3) |
7.21 |
5.24 (± 2.85) |
4.93 (1.85−10.92) |
4.60 |
|
Cd (ng/m3) |
3.74 |
1.72 (± 0.76) |
1.76 (0.56−3.45) |
0.82 |
|
Pb (ng/m3) |
419.38 |
44.32 (± 16.16) |
41.95 (19.03−90.86) |
48.51 |
|
Cr (ng/m3) |
23.00 |
10.41 (± 3.78) |
9.73 (6.85−21.01) |
13.17 |
|
Mn (ng/m3) |
131.33 |
48.89 (± 15.62) |
47.30 (29.87−85.38) |
51.57 |
|
V (ng/m3) |
9.95 |
5.6 (±1.93) |
5.60 (3.06−11.50) |
23.36 |
|
Ni (ng/m3) |
47.87 |
5.27 (± 2.12 |
4.90 (3.54−10.26) |
15.27 |
|
Zn (ng/m3) |
758.01 |
170. (± 41) |
163.18 (43.51−357.35) |
214.39 |
|
Cu (ng/m3) |
287.99 |
66.81 (± 101.45) |
68.61 (8.17−368.75) |
59.43 |
|
Fe (ng/m3) |
4771.38 |
1763.88 (± 2.85) |
1503.17 (986.84−4010.48) |
1136.26 |
|
TSP (μg/m3) |
457.42 |
183.78 (± 104.83) |
155.42 (106.71−411.34) |
128.47 |
|
As (ng/m3) |
12.67 |
6.39 (± 2.64) |
7.33 (2.88−11.43) |
5.07 |
|
Cd (ng/m3) |
6.19 |
2.04 (± 0.67) |
2.24 (0.88−3.37) |
0.84 |
|
Pb (ng/m3) |
1224.95 |
54.28 (± 17.59) |
56.18 (24.55−92.29) |
67.78 |
|
Cr (ng/m3) |
36.77 |
17.81 (± 5.63) |
18.42 (11.51−36.01) |
19.73 |
|
Mn (ng/m3) |
295.21 |
86.06 (± 44.43) |
77.89 (45.88−196.36) |
61.25 |
|
V (ng/m3) |
15.60 |
7.66 (± 4.46) |
6.72 (4.33−23.26) |
21.74 |
|
Ni (ng/m3) |
126.30 |
8.72 (± 3.7) |
9.04 (4.46−18.09) |
18.10 |
|
Zn (ng/m3) |
1636.70 |
250.29 (± 104.12) |
248.36 (113.33−505.69) |
266.25 |
|
Cu (ng/m3) |
1126.66 |
70.11 (± 77.67) |
75.95 (11.45−315.66) |
75.75 |
|
Fe (ng/m3) |
15309.03 |
1507.04 (± 2.64) |
2623.64 (1470.36−11297.02) |
1727.61 |
SD: standard deviation
Comment (5): At the description of the table, you can write a note describing ± SD as the standard deviation.
Response: Thanks for your advice. In the revised manuscript, a footnote was added in Table 1 and revised as follows:
SD: standard deviation.
Comment (6): Use either TSP or PM10-100 for uniformity in the text.
Response: Thanks for your advice. TSP and PM10-100 are particulates in two particle size ranges. TSP is particulate matter with an aerodynamic diameter ≤ 100 μm, while PM10-100 is defined as particulate matter with an aerodynamic diameter between 10 to 100 μm. Thus, TSP and PM10-100 are not interchangeable. However, one of the TSP was removed to avoid confusion.
(P13, lines 440)
Metals distributed in the PM2.5-10 samples made up about 14.72%−39.81% of the TSP level.
